

# Genetic diversity of Microsporidia in the circulatory system of endemic amphipods from different locations and depths of ancient Lake Baikal

Mariya Dimova[1], Ekaterina Madyarova[1,2], Anton Gurkov[1,2], Polina Drozdova[1], Yulia Lubyaga[1,2], Elizaveta Kondrateva[1], Renat Adelshin[1,3] and Maxim Timofeyev[1]

[1] Irkutsk State University, Irkutsk, Russia
[2] Baikal Research Centre, Irkutsk, Russia
[3] Irkutsk Anti-Plague Research Institute of Siberia and Far East, Irkutsk, Russia

Corresponding author
Maxim Timofeyev,
m.a.timofeyev@gmail.com

## ABSTRACT

Endemic amphipods (Amphipoda, Crustacea) of the most ancient and large freshwater Lake Baikal (Siberia, Russia) are a highly diverse group comprising > 15% of all known species of continental amphipods. The extensive endemic biodiversity of Baikal amphipods provides the unique opportunity to study interactions and possible coevolution of this group and their parasites, such as Microsporidia. In this study, we investigated microsporidian diversity in the circulatory system of 22 endemic species of amphipods inhabiting littoral, sublittoral and deep-water zones in all three basins of Lake Baikal. Using molecular genetic techniques, we found microsporidian DNA in two littoral (*Eulimnogammarus verrucosus*, *Eulimnogammarus cyaneus*), two littoral/-sublittoral (*Pallasea cancellus*, *Eulimnogammarus marituji*) and two sublittoral/deep-water (*Acanthogammarus lappaceus longispinus*, *Acanthogammarus victorii maculosus*) endemic species. Twenty sequences of the small subunit ribosomal (SSU) rDNA were obtained from the haemolymph of the six endemic amphipod species sampled from 0–60 m depths at the Southern Lake Baikal's basin (only the Western shore) and at the Central Baikal. They form clusters with similarity to *Enterocytospora*, *Cucumispora*, *Dictyocoela*, and several unassigned Microsporidia sequences, respectively. Our sequence data show similarity to previously identified microsporidian DNA from inhabitants of both Lake Baikal and other water reservoirs. The results of our study suggest that the genetic diversity of Microsporidia in haemolymph of endemic amphipods from Lake Baikal does not correlate with host species, geographic location or depth factors but is homogeneously diverse.

## INTRODUCTION

Lake Baikal is the deepest (1,643 m) and oldest lake in the world (25–30 million years); it is the largest (by volume: 23,000 km$^3$) reservoir of clean fresh water, containing approximately 20% of the available world's liquid fresh water (*Martens, 1997*; *Yoshii, 1999*). Lake Baikal

has specific abiotic characteristics that distinguish it from all other freshwater bodies in the world: high oxygen content throughout the entire water column, stable low water temperatures with long seasonal ice coverage on the lake's surface, and super-oligotrophic conditions. The environmental conditions of the lake's open-water and deep-water zones have remained close to their current state for the last 2–4 million years (*Kozhova & Izmest'eva, 1998*). As a unique ecosystem with exceptionally high degrees of biological diversity and endemism, Lake Baikal was designated a UNESCO World Heritage Site. To date, 2,595 animal species from Lake Baikal have been described, approximately 80% of which are endemic (*Timoshkin, 2001*).

Among macroinvertebrates, the highest number of species in the lake is presented by amphipods. The diversity of amphipods in Lake Baikal is very high: currently, 276 species and 78 subspecies (over 10% of the total species diversity of fauna of Lake Baikal; over 15% of all recognized amphipod species from fresh or inland waters world-wide), grouped in seven families and 41 genera, have been found in the lake (*Väinölä et al., 2008*; *Takhteev, Berezina & Sidorov, 2015*). Amphipods inhabit all depths of Baikal, including littoral (0–20 m), sublittoral (20–70 m), supra-abyssal (70–250 m) and abyssal (250–1,640 m); the last two depths are typically grouped as a deep-water zone (*Kozhov, 1962*). The age and geographical isolation of this group create excellent opportunities for exploring the diversity and evolution of host-parasite relationships.

Microsporidia are a diverse phylum of eukaryotic parasites, which are a sister clade to the Fungi kingdom and infect a wide range of hosts from invertebrates to humans (*Keeling & Fast, 2002*; *Smith, 2009*; *Issi et al., 2012*; *Gismondi et al., 2012*; *Stentiford et al., 2013*; *Bojko et al., 2017b*). The number of described species of Microsporidia is more than 1,300, and the various species belong to approximately 187 genera, 50 of which are found in aquatic arthropods (*Stentiford et al., 2013*; *Weiss & Becnel, 2014*). Some microsporidian species may influence the sex ratio of arthropod populations (*Terry, Smith & Dunn, 1998*; *Ironside et al., 2003*; *Ironside & Alexander, 2015*), the behaviour of the host, and the host population dynamics (*Dunn & Smith, 2001*). For example, a selective sexual behaviour is described for amphipod species *Gammarus duebeni* Liljeborg, 1852. The males of this species infected by Microsporidia are only able to hold down infected females for breeding (*Dunn, Hogg & Hatcher, 2006*). Microsporidia can be transmitted horizontally, vertically or both ways (*Terry et al., 2004*; *Smith, 2009*). They have been found to infect muscles, gonads, intestinal walls, hepatopancreas, haemocytes, and other organs (*Kelly & Anthony, 1979*; *Weiss & Becnel, 2014*).

Despite the fact that amphipods endemic to Lake Baikal offer great possibility to study the basic principles of coevolution of the host-parasite system and distribution of parasites among evolutionary close species, only a few papers have been published on the microsporidian diversity in the lake's fauna. The study of Microsporidia in Baikal amphipods began in 1967 when *Nosema kozhovi* was found in intestinal epithelium of *Brandtia latissima lata* (Dyb., 1874) (*Lipa, 1967*). The first molecular genetic studies of Microsporidia were performed in the 21st century. Six endemic and one cosmopolitan microsporidian species (*Dictyocoela duebenum*) were detected in the Baikal amphipod *Gmelinoides fasciatus* (Stebbing, 1899) (*Kuzmenkova, Sherbakov & Smith, 2008*), and 100

new gene isolates of Microsporidia in 31 species of amphipods of Lake Baikal were discovered by molecular methods (*Smith et al., 2008*). A recent study provided the first glimpse at horizontal distribution and exchange of Microsporidia between Baikal littoral and non-Baikal amphipods, which indicated frequent introductions of the parasites into Lake Baikal ecosystem and high homogeneity of Microsporidia between species of endemic hosts in the coastal zone (*Ironside & Wilkinson, 2017*).

However, most previous studies utilized the whole individuals or soft amphipod tissues for amplification of microsporidian DNA, which does not rule out the possibility of contamination. To avoid this problem, our group recently conducted a study using molecular genetic techniques (SSU rDNA sequencing) to detect Microsporidia in circulatory system of several endemic amphipod species from Lake Baikal (*Madyarova et al., 2015*). Searching directly in the haemolymph minimizes the possibility of identifying SSU rDNA of Microsporidia located on exoskeleton, in the gut lumen or inside parasites of the amphipods and can guarantee that the found Microsporidia are parasitic directly to the amphipod species studied. Moreover, using haemolymph provides a unified way of treating amphipods of very different sizes, including small species and young individuals.

Spatial variations, especially vertical distribution, in biodiversity of Microsporidia in Lake Baikal amphipods remain poorly investigated. Thus, the aim of the current study was to search for diversity of Microsporidia in the haemolymph of endemic amphipods inhabiting littoral, sublittoral and deep-water zones of the lake in all three basins of Baikal: Southern, Central and Northern.

## MATERIALS AND METHODS

### Sampling and location

To study microsporidian biodiversity, the haemolymph of 22 amphipod species from 11 points of Lake Baikal was used (Table S1). The following seven species were collected at depths of 0–1 m using a hand net in the period from 2011 to 2016: *Eulimnogammarus verrucosus*, *E. cyaneus*, *E. marituji*, *E. vitatus*, *E. maaki*, *E. viridis olivaceus*, *Pallasea cancelus*. The other amphipod species were sampled from depths of 20–200 m: *Acanthogammarus victorii maculosus*, *A. flavus*, *A. lappaceus longispinus*, *A. godlewskii*, *A. brevispinus*, *A. reicherii*, *Carinurus belkini*, *Garjajewia cabanisi*, *Propachygammarus maximus*, *Ommatogammarus carneolus melanophthalmus*, *Parapallasea lagovskii*, *P. puzyllii*, *P. brandtia*, *P. kesslerii*, *P. grubii*. The latter group of species was collected during the expeditions in 2014 and 2016 using a trawl, dredge and deep-water traps with rotten fish installed at a depth of 200 m. In this work, only benthic amphipod species were studied. The chosen amphipod species are not endangered or protected.

Species determination was performed according to the identification key (*Bazikalova, 1945*; *Takhteev & Didorenko, 2015*).

The collections cover all three of Lake Baikal's basins: the Southern (Listvyanka, Bolshie Koty, Port Baikal), the Central (Olkhon Island, Kharauz, Ushkany Islands) and the Northern (Kotelnikovskiy Cap), including both the Western (Listvyanka, Bolshie Koty, Port Baikal) and Eastern (Kharauz) shores (Fig. 1). Around Olkhon Island, five sampling points
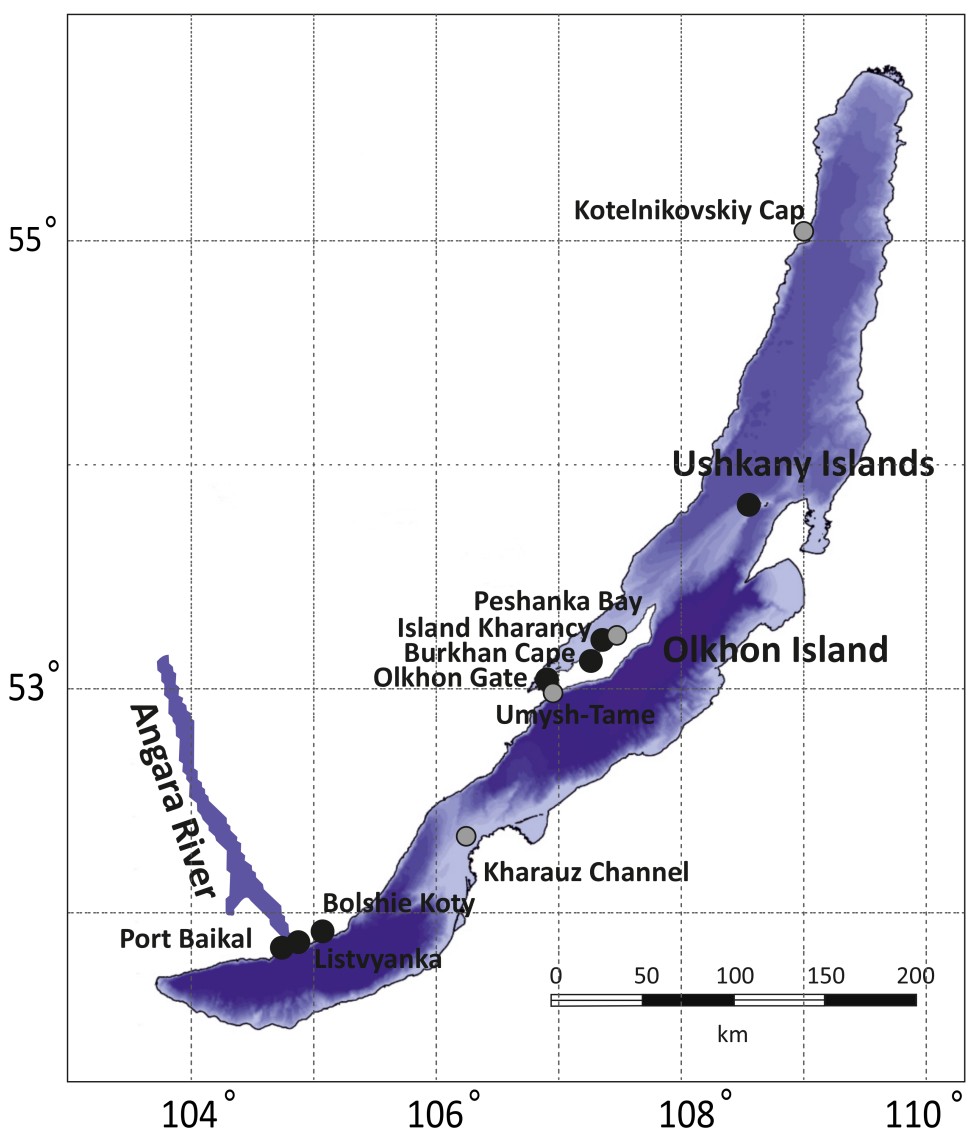

**Figure 1** **Sampling sites for amphipod collection in Lake Baikal.** Grey dots correspond to sites where no infected amphipods were found, while black dots mark places where microsporidian DNA was identified in amphipod haemolymph. The bathymetric map of Lake Baikal was taken from http://users.ugent.be/~mdbatist/intas/intas.htm and modified with CorelDRAW.

were screened: Burkhan Cape, Cape Umysh-Tame, Olkhon Gate, Kharantsy Island and Peschanka Bay. The Lake Baikal's basins and the Angara River act as natural geographic barriers for benthic amphipods (*Mashiko, 2000*; *Gomanenko et al., 2005*). No specific permissions were required for sampling of amphipods in these locations.

## Haemolymph extraction

Haemolymph extraction was performed from live amphipods in a laboratory using a glass capillary. Each sample subsequently used for DNA extraction (volumes ranging from 50 to 100 μl) contained haemolymph collected from one or pooled from several individuals

of the same species. The number of individuals per sample depended on the size of the amphipods (1–50 individuals; see Table S1 for details). Isolated haemolymph samples were stored in liquid nitrogen.

## DNA extraction and PCR

Total DNA extraction from individual or pooled haemolymph was performed using the "Riboprep" reagent kit (AmpliSens, Moscow, Russia). The concentration and purity of the isolated DNA was determined using the UV spectrophotometer UNICO 2802 (UNICO, Franksville, WI, USA).

The small subunit of ribosomal DNA (SSU rDNA) of the Microsporidia was selected as a molecular phylogenetic marker. Amplification was performed in the nested PCR with universal primers for Microsporidia: V1f: 5′-CACCAGGTTGATTCTGCCTGAC-3′ (*Weiss et al., 1994*); 1342r: 5′-ACGGGCGGTGTGTACAAAGAACAG-3′ (*McClymont et al., 2005*); 18sf: 5′-GTTGATTCTGCCTGACGT-3′ (*Baker et al., 1995*); and 981r: 5′-TGGTAAGCTGTCCCGCGTTGAGTC-3′ (*MacNeil et al., 2003*). Each PCR was performed in a Gradient Thermocycler (Biometra, Göttingen, Germany) in a volume of 25 μl and contained 1X PCR buffer, 6.25 pmol of each dNTP, 2.5 U SynTaq DNA polymerase, 6.25 pmol $MgCl_2$, 10 pmol primers, 3 μl of DNA template, and deionized water. The conditions for the first round were 95 °C for 5 min, followed by 40 cycles of 95 °C for 30 s, 55 °C for 30 s, and 72 °C for 90 s, and then 72 °C for 7 min. Conditions for the second round of PCR were 95 °C for 5 min, followed by 35 cycles of 95 °C for 30 s, 52 °C for 30 sec, and 72 °C for 1 min, followed by 72 °C for 7 min.

The PCR products of the second round were visualized in 1% agarose gels, and products of the expected size were excised from the gel and purified by ethanol and sodium acetate precipitation (*Sambrook, Fritsch & Maniatis, 1989*). Sequencing of the amplified DNA fragments was performed using Genetic Analyzer 3500 xL (Applied Biosystems, Tokyo, Japan) with the BigDye Terminator Cycle Sequencing kit v.3.1 (Applied Biosystems, Foster City, CA, USA).

## Nucleotide sequence analysis, phylogenetic reconstruction and statistical analysis

The obtained sequences were identified with NCBI nucleotide BLAST with default parameters against the nt database (last accessed 2018/05/17). The best B hit belonging to a sample identified in other studies was chosen to describe the samples (Table S2).

The sequences were aligned with the MAFFT 7.397 E-INS-i strategy (*Katoh, Asimenos & Toh, 2009*) and then trimmed with Gblocks (*Castresana, 2000*) accessed through the Phylogeny.fr web interface (*Dereeper et al., 2008*) with all options for less stringent selection checked. The resulting alignment (File S1) contained 514 bases. Nucleotide substitution models were evaluated with jModelTest 2.1.10 (*Guindon & Gascuel, 2003*; *Darriba et al., 2012*). The best model according to Bayesian information criterion (BIC) score was TrN+G with GTR+G falling very close behind. As the formed model was not available in RAxML, GTR+G was used for all calculations. The phylogenetic tree was reconstructed with BEAST 2.5 (*Bouckaert et al., 2014*) for 10,000,000 generations with sampling every 1,000th tree.

After this run, ESS value was close to 150, indicating acceptable convergence. In addition, we reconstructed the tree with RAxML 8.2.11 (*Stamatakis, 2014*) under the GTRGAMMA model and 5,000 rapid boostrap generations. RAxML bootstrap results were reanalysed with the tree obtained with BEAST to get comparable support rates for the nodes. Raw trees obtained with BEAST and RAxML are shown in Tree S1 and Tree S2, respectively. The results were visualized in FigTree 1.4.2 (http://tree.bio.ed.ac.uk/software/figtree/) and Dendroscope, and Fig. 2 was prepared with the ggtree package (*Yu et al., 2017*) for R (*R Core Team, 2017*) and Inkscape (https://inkscape.org/en/).

Simpson's diversity index was calculated using the diversity function (option *index* set to *invsimpson*) of the vegan package (*Dixon, 2003*) for R. Fisher's exact test was implemented with the stats package for R.

## RESULTS

We screened the haemolymph of 22 species (Table S1) of endemic amphipods from Lake Baikal and found microsporidian DNA in two littoral (*E. verrucosus*, *E. cyaneus*), two littoral/sublittoral (*P. cancellus*, *E. marituji*) and two sublittoral/deep-water species (*A. lappaceus longispinus*, *A. victorii maculosus*). A total of over 1,000 individual amphipods were analysed (Table S1). Twenty nucleotide sequences of SSU rDNA belonging to the *Cucumispora*, *Dictyocoela*, *Enterocytospora*-like groups and unassigned Microsporidia were obtained from the haemolymph of the six endemic amphipod species sampled from depths of 0–60 m at Southern Lake Baikal's basin (only the Western shore) and at Central Baikal (Table S2). Only one microsporidian isolate has been identified in each positive pooled haemolymph sample. The phylogenetic tree (Fig. 2) reconstructed using these and some other published microsporidian sequences (File S1) shows clustering of the DNA identified in the current study to four groups.

Three microsporidian DNA sequences were found in the haemolymph of *E. cyaneus* (sampled at 0–1 m in the Southern Baikal), and one was found in *A. victorii maculosus* (sampled at 53–60 m in the Central Baikal). These four sequences are presented as a single cluster on the phylogenetic tree marked as *Cucumispora*-like (C) and are similar to taxonomically defined *Cucumispora* isolates (*Ovcharenko et al., 2010*; *Bojko et al., 2015*; *Bojko et al., 2017b*) as well as microsporidian sequences found earlier in some other endemic amphipods of Lake Baikal, such as *P. cancellus* (KM977842, KM977843, KM977844, KM977845, KM977846; *Madyarova et al., 2015*), *Acanthogammarus victorii* (FJ756173), *Garjajewia cabanisii* (FJ755959) and *Pallaseopsis kessleri* (FJ756022).

The other microsporidian DNA from one haemolymph sample of *P. cancellus*, three samples of *E. verrucosus* (both sampled at 0–1 m in the Southern Baikal) and two of *A. lappaceus longispinus* (sampled at 25–35 m in the Central Baikal) were clustered with those of Microsporidia of the recently defined (*Bacela-Spychalska et al., 2018*) genus *Dictyocoela* (D), namely *D. duebenum* (*Grabner et al., 2015*) and *D. berillonum*. Representatives of *Dictyocoela* were found earlier in an endemic Baikal amphipod *A. lappaceus* (KM977839; *Madyarova et al., 2015*).
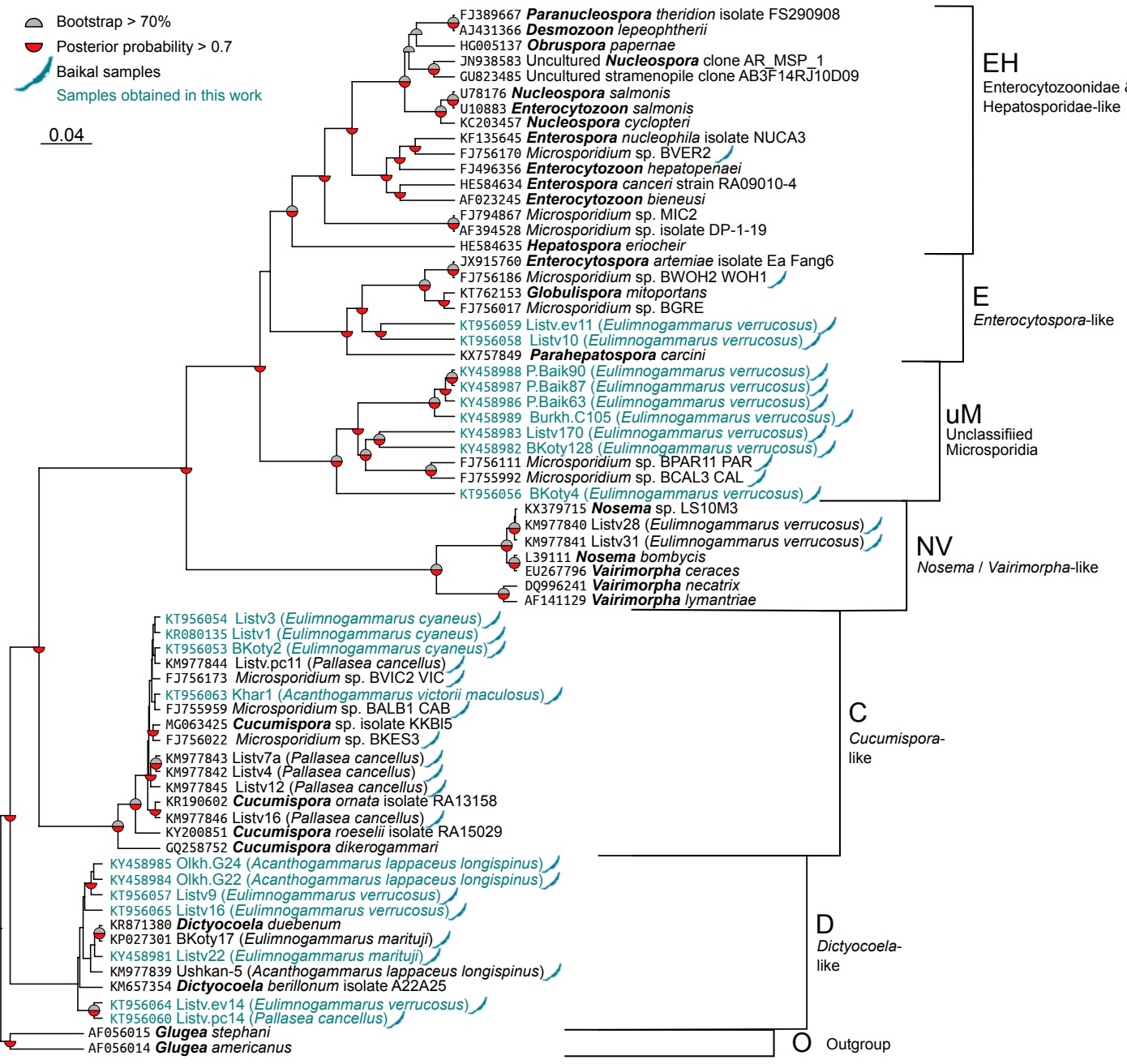

**Figure 2** **Bayesian phylogenetic tree of Microsporidia identified in amphipods of Lake Baikal and some other crustaceans.** The tree contains the following clusters: Enterocytozoonidae&Hepatosporidae-like (EH), *Enterocytospora-* (E), *Cucumispora-* (C), *Dictyocoela-* (D), *Nosema-Vairimorpha-* like (NV), and some unclassified Microsporidia (uM). Branch lengths are drawn to scale. Grey semi-circles mark nodes with bootstrap support >70%, and red semi-circles mark those with posterior probability >0.7. Baikal shapes mark microsporidian sequences amplified from Lake Baikal amphipods, and samples obtained in this work are in blue font. The alignment used to build this tree is presented in File S1.

Two more sequences amplified from *Eulimnogammarus verrucosus* haemolymph clustered with other isolates of the *Enterocytospora*-like clade (E), such as *Enterocytospora artemiae* (JX915760; *Rode et al., 2013a*), *Globulispora mitoportans* (KT762153; *Vávra et al., 2016*) and recently described *Parahepatospora carcini* (KX757849; *Bojko et al., 2017a*). A similar sequence was also found in an endemic Baikal amphipod *Micruropus wahlii* (FJ756186).

Finally, some isolates detected in the littoral species *E. verrucosus* (sampled at 0–1 m) from four different habitats, Listvyanka, Bolshie Koty, Port Baikal (Southern Baikal) and Burkhan Cape (Central Baikal), clustered with other microsporidian sequences previously amplified from Baikal endemic amphipods (FJ756111 from *Dorogostaiskia parasitica* and FJ755992 from *Odontogammarus calcaratus*). This cluster is designated as unclassified Microsporidia (uM).

As we have data for seven different locations, we have calculated Simpson's diversity index for the groups of amphipods defined according to the phylogenetic tree and these locations (Fig. S1). Our samples are not big enough to draw any conclusions, as we quite predictably saw higher diversity in locations with higher numbers of analysed pools, but further studies may help clarify this issue.

## DISCUSSION

The microsporidian DNA sequences found in the haemolymph of amphipods from Lake Baikal during this study belong to four clusters: *Cucumispora*-, *Dictyocoela*-, *Enterocytospora*-like, as well as some unclassified Microsporidia. Some other sequences belonging to the *Cucumispora*-like and *Dictyocoela*-like clusters, as well as to unclassified Microsporidia were also identified in our previous work (*Madyarova et al., 2015*). Apart from these similarities, there were also some differences, as in the earlier work we also identified representatives of the genus *Nosema* (N) in Baikal amphipods *E. verrucosus* from Listvyanka (*Madyarova et al., 2015*), but in the current study, these microsporidian species were not found at all.

Moreover, in winter 2013–2014 (Table S1) *E. verrucosus* from Listvyanka showed relatively high infection (∼30% of pools) by Microsporidia of groups *Nosema/Vairimorpha*- (NV), *Dictyocoela*- (D), *Enterocytospora*- like (E) and unclassified Microsporidia (uM), while in summer 2016 only one infected pool (∼3%) was identified in this location and corresponded to an unclassified Microsporidia (uM). We have compared the largest samples with exact Fisher's test (5 out of 18 *vs.* 1 out of 33) and found that the difference was statistically significant ($p = 0.017$). Sampling of Microsporidia during summer and autumn (2015 and 2016) at other points (Port Baikal, Bolshie Koty and Burkhan Cape) also demonstrated relatively low infection (no more than 13% of pools) only by unclassified Microsporidia. It is interesting to note that two highest infection rates belonged to pools sampled in winter, the reproduction season of *E. verrucosus* (*Bazikalova, 1941*). Generally, by comparing the percentage of infected haemolymph pools of different amphipod species from chosen sampling points (Table S1), we can conclude that the proportion of infected individuals can vary significantly.

The species diversity of the obtained samples is also interesting in the context of worldwide diversity of microsporidian parasites of amphipods. Microsporidian SSU rDNA nucleotide sequences similar to the C group in our analysis (Fig. 2) were also found in *Gammarus chevreuxi* Sexton, 1913 from the Avon River in the UK (AJ438962) (*Terry et al., 2004*). The identified sequences of the D group are similar to the *Dictyocoela duebenum* isolate 775 (FN434091), which was previously found in *Gammarus duebeni* from Iceland (*Krebes et al., 2010*), and the *Dictyocoela muelleri* isolate (AJ438955) found in *Gammarus duebeni celticus* from Ireland (*Terry et al., 2004*). The *Enterocytospora*-like sequences are similar to the sequences of *Enterocytospora artemiae* found in *Artemia franciscana monica* from Mono Lake, USA (*Rode et al., 2013b*). These results may suggest that microsporidian distribution in Baikal amphipods is similar to that in amphipods and other crustaceans from other water bodies.

Concerning the spatial and host distribution of Microsporidia in Baikal amphipods, we should note that similar and identical sequences of microsporidian DNA were found in amphipods inhabiting different depths and basins of the lake. Two microsporidian sequences from *P. cancellus* (KM977844 and KM977846) were identical to sequences from completely different amphipod species *G. cabanisi* (FJ755959) and *D. parasitica* (FJ756113) (Table S2 ; Fig. 2). Almost identical DNA of Microsporidia that belong to the group C were found at sampling depths 0–1 m of Southern Baikal (KT956054, KR080135, KT956053 from *E. cyaneus* and KM977844 from *P. cancellus*) and depths of 54-60 m of Central Baikal (KT956063 from *A. victorii maculosus*). Similar sequences of Microsporidia of the group D are found at depths of 0–1 m (*E. marituji*, *E. verrucosus* and *P. cancellus* from the Southern Baikal) and at 25–35 m (*A. lappaceus longispinus* from the Central Baikal). Additionally, amphipods of species *E. verrucosus* from Southern and Central Baikal were infected by microsporidia with almost identical SSU rDNA belonging to some unclassified Microsporidia (uM). So, similar microsporidian isolates infect different hosts at different depths and locations. There are several possible explanations for this fact. Amphipods are omnivorous animals and can feed on other amphipods of different or the same species. Moreover, Baikal amphipods are characterized by high vertical and horizontal mobility: a significant part of the analysed species (*E. marituji*, *P. cancellus*, *A. lappaceus longispinus* and others) can be found in a range of depths of at least several dozen meters and can participate in nocturnal vertical migrations observed for many benthic animals in Baikal (*Bazikalova, 1945*; *Karnaukhov et al., 2016*); also, in Lake Baikal there are eurybathic scavenger amphipods (like *Ommatogammarus flavus* and *O. albinus*), found in depth ranges of several hundred meters (*Bazikalova, 1945*), which may serve as additional "carriers" of infections in the benthic community.

Thus, the results of our study do not suggest any link between the genetic diversity of Microsporidia found in circulatory system of endemic amphipods from Lake Baikal and host species, geographic location or depth. This result corroborates previous studies (*Madyarova et al., 2015*; *Ironside & Wilkinson, 2017*).

## CONCLUSIONS

The sequence isolates that we have identified during our study belong to four clusters and show similarity to previously identified microsporidian DNA from inhabitants of both Lake Baikal and other water reservoirs. No specificity in distribution of Microsporidia was identified depending on host amphipod species and their spatial relations, which may indicate high connectivity of individuals within species and between species of Lake Baikal amphipods. However, the search for Microsporidia in amphipods at greater depths (200–1,600 m) is still necessary to confirm the observed homogeneity in distribution of these parasites.

## ACKNOWLEDGEMENTS

We would like to thank Dr Denis Axenov-Gribanov, Dr Daria Bedulina, Prof. Vadim Takhteev and the crew of the research vessel ''Prof. M. Kozhov'' for assistance in the selection and fixation of samples, identifying species of amphipods and their valuable advice. We are extremely grateful to Dr Jamie Bojko and an anonymous peer reviewer for their helpful comments on the manuscript.

### Funding

This work was supported by the Russian Science Foundation (#17-14-01063), the Ministry of Education and Science of Russia («Goszadanie»: #6.1387.2017/4.6, #6.9654.2017/8.9) and the Russian Foundation for Basic Research (#15-29-01003, #17-44-388067, #18-34-00294). The funders had no role in study design, data collection and analysis, decision to publish, or preparation of the manuscript.

### Grant Disclosures

The following grant information was disclosed by the authors:
Russian Science Foundation: #17-14-01063.
The Ministry of Education and Science of Russia («Goszadanie»): #6.1387.2017/4.6, # 6.9654.2017/8.9.
Russian Foundation for Basic Research: #15-29-01003, #17-44-388067, #18-34-00294.

### Competing Interests

The authors declare there are no competing interests.

### Author Contributions

- Mariya Dimova and Ekaterina Madyarova performed the experiments, analyzed the data, prepared figures and/or tables, authored or reviewed drafts of the paper, approved the final draft.
- Anton Gurkov and Polina Drozdova analyzed the data, prepared figures and/or tables, authored or reviewed drafts of the paper, approved the final draft.

- Yulia Lubyaga and Elizaveta Kondrateva performed the experiments, approved the final draft.
- Renat Adelshin conceived and designed the experiments, analyzed the data, contributed reagents/materials/analysis tools, authored or reviewed drafts of the paper, approved the final draft.
- Maxim Timofeyev conceived and designed the experiments, contributed reagents/-materials/analysis tools, authored or reviewed drafts of the paper, approved the final draft.

## DNA Deposition

The following information was supplied regarding the deposition of DNA sequences:

The microsporidia sequences described here are accessible via GenBank accession numbers: KT956054, KT956053, KR080135, KT956063, KT956057, KT956065, KT956064, KT956058, KT956059, KY458983, KY458982, KT956056, KY458986, KY458987, KY458988, KY458989, KT956060, KY458984, KY458985, KY458981.

## Data Availability

The raw sequence alignment is provided in the Supplemental Files.

## Supplemental Information

Supplemental information for this article can be found online at http://dx.doi.org/10.7717/peerj.5329#supplemental-information.

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
