# Peer review of "Genetic diversity of Microsporidia in the circulatory system of endemic amphipods from different locations and depths of ancient Lake Baikal"

_PeerJ, doi:10.7717/peerj.5329_

## Round 0.1 · original submission · Major Revisions

Dear Authors,

The reviews have now been received on your manuscript and both reviewers see merit in the research submitted. One reviewer has highlighted that more detail from your analysis is required to support your conclusions hence I have recommended major revisions to give you more time to address their concerns.

·

Basic reporting

English – Grammatical errors are present, but should be picked up during editing. I have identified some myself.
References – Some key references are missing, but the style is fine in most cases. The authors sometimes swap between year ascending or year increasing.
Structure – The structure of the manuscript is fine, but sections on statistical analyses are missing, and the phylogenetics needs much more information.
Hypotheses – The hypothesis is interesting, however much work is needed to draw scientific conclusions from the data.

Experimental design

Original – Yes, the work is original and of interest to the scientific community.
Research question – The research question is clear, however it is not fully explored and there is much needed from the authors to make this a publishable manuscript.
Technical and ethical standard – Ethical standard is fine. Technical work seems fine, but more detail is needed in the methods.
Methods – More detail needed. Specifics are listed below in general comments.

Validity of the findings

Impact – Good. Of interest to microsporidian researchers and taxonomists as well as ecologists.
Data – The phylogenetics needs inclusion of taxonomically defined species, such as: Cucumispora dikerogammari; Cucumispora ornata; Cucumispora roeselii; Nosema bombycis; Variamorpha species (several); and members of the Enterocytospora-like clade (clearly displayed in Bojko et al. 2016 and Vavra et al. 2016), including Hepatospora eriocheir and Parahepatospora carcini (if gene sequence data crosses over and is able to align) and members of the Entercoytozoonidae. Please use the Vossbrinck and Debrunner-Vossbrinck 2005 paper on the 5 clades of Microsporidia, defined using molecular phylogeny, where appropriate. The diversity data from your paper needs to be properly analysed. No analysis is present. No significance data is present for DNA similarity: comparison coverage?; e-value?; similarity? All of these are not present and need to be included – All are output from BLAST (NCBI).
Conclusion – Speculative, but the data are not fully analysed and the authors have much to do before this can be properly drawn.
Speculation – Yes. The data are not properly analysed (or at all?). Once this is done I will reassess this when I review it for the second time.

Additional comments

Title: It should be Microsporidia. Capital M.
Line 25: change to ‘microsporidian’
Line 31: When using the phylum name a capital M is needed for Microsporidia. Please change this throughout the manuscript.
Line 32: ‘Microsporidium’ is not a genus within the Microsporidia, it is in fact a holding genus for unassigned genetic data. I suggest rewording this to “ The 20 nucleotide sequences of small subunit ribosomal (SSU) rDNA (681bp) showed similarity to the Cucumispora, the unofficial genus Dictyocoela, and several unassigned microsporidia sequences. These were obtained from…”
Line 34: Alter the text to “Our sequence data show similarity to…”
Line 37: In the haemolymph
Line 36-38: I think you should re-word this to: “The results of our study indicate that microsporidian diversity in the endemic amphipods of Lake Baikal does not corelate with host, geographic location, or depth factors, but remains homogenously diverse throughout.”
Line 41: This initial description of microsporidian taxonomy is misleading. I suggest replacing this first sentence with a more accurate determination of where Microsporidia sit phylogenetically relative to other organisms. I suggest reading the introduction of Bojko et al. 2016, which describes parahepatospora; Stentiford et al. 2015, which describes Paradoxium; and data surrounding the Opisthosporidia, which I believe was published in a 2014 paper by Karpov et al. – Those papers will give you a good foundation for your introductory sentence.
Line 45: The reference is Stentiford et al. 2013.
Line 46: Microsporidian can influence host sex characteristics. Not all do this. The referencing here reverses direction.
Line 47: American English is used? I think the journal requires British English.
Line 51-52: I would just state horizontally or vertically transmitted. Horizontal transmission is not just through food.
Line 52-53: They have been noted to infect…
Line 55: I think you should start your introduction with this line. Then once you get to line 75 add in your paragraph on Microsporidia. This will help your manuscript flow better.
Line 77: microsporidian
Line 83: You should call these ‘gene isolates’ not species. Most sequence data linking to the Microsporidia are yet to have full taxonomic description.
Line 132: This is slightly confusing. Were the haemolymph samples from several amphipods of the same species pooled into a single extraction and PCR? Please be explicit here or refer to where this specific data can be located.
Line 136: DNA extraction, not isolation.
Line 146: This is a lot of dNTP… But if it is correct then I suppose the excess would not have inhibited the reaction in any way. Please check the calculation to be sure.
Line 157: This should be titled “Nucleotide sequence analysis and phylogenetics”. You need to describe your use of BLAST (NCBI), if that is what you used to check similarity to other sequence isolates. Your phylogenetics has a lot that isn’t mentioned. I need to know which algorithm you used to test if your analysis is reasonable… was it Tamura92/Jukes and Cantor/ etc? I need to know if the sequences were trimmed. What the gap costs were. As well as a variety of other information. You can have a look at other studies that display this information to see what is missing. The previously suggested papers will also help with this: Bojko et al. 2016; Stentiford et al. 2015 – Microsporidia descriptions.
Methods: Do you not have any statistical analysis of your data? You should be able to provide a basic analysis of whether there is significant diversity changes between the depths and locations. Perhaps use Simpsons Diversity Index and then compare using an ANOVA, if the data are normal.
Line 165: remove the “have”.
Line 168-169: Refer back to my comment above about these “genera” and adjust your statement accordingly. Microsporidium = holding genus. Dictyocoela = unofficial genus. Cucumispora = fully characterised genus.
Line 171-174: This is unclear. Do the authors mean that only one genetic isolate was detected from each positive sample? If so, this should be stated clearly.
Figure 2: The description of this figure needs altering. Cucumispora was described by Ovcharenko et al. 2010, followed by the second description of a species in this genus by Bojko et al. 2015, followed by a third description in Bojko et al. 2017, who also developed a phylogeny exploring the additional sequence isolates that likely belong to the Cucumispora. You should utilise these three references in combination when explaining your phylogeny, because the background data will help you describe the position of your new isolates.
Figure 1: You need to list the tool you used to make your map. You should also show all the places you collected and tested amphipods for Microsporidia. This negative data is also useful. Perhaps list the sites with Microsporidia in red or with a red dot.
Figure 2: This figure is missing almost all of the taxonomically identified species sequence isolates. For example, you have not included Cucumispora dikerogammari (see Ovcharenko et al. 2010), Cucumispora ornata (see Bojko et al. 2015) or the most recent Cucumispora roeselii (see Bojko et al. 2017). For the Dictyocoela, you have not provided the sequence of D. berillonum or many others. Your section ‘M’ should not be called Microsporidium. This group is explored in Bojko et al. 2016 and you should include the same isolates, such as Globulispora mitoportens, Parahepatospora carcini, Hepatospora eriocheir, and many many other fully described species. You cannot provide a readable tree unless you are using these known species as anchors to view your phylogeny. In essence, your phylogenetic tree needs completely re-doing with the inclusion of many known Microsporidia for which sequence data is available. I also suggest using a combination of Maximum Likelihood and Bayesian methods, or Neighbour Joining to give two perspectives on the data. You can also use Bojko et al. 2016 to help you to see how this should be displayed.
Line 172: This is a phylogenetic tree. Dendrogram, cladogram etc are confusing and I would stick to this terminology throughout.
Line 182: You should include these sequences. It will make your phylogenetic tree much better and informative.
Line 188: From the unofficial genus, Dictyocoela. This genus is awaiting full taxonomic confirmation. Also, “found earlier”.
Line 192: remove the macronuclear
Line 194: This is not a cluster of Microsporidium, essentially here you are detecting isolates that clade with the Enterocytospora-like clade, the Enterocytozoon genus and the Enterocytozoonidae. When you include many more of the described Microsporidia in your tree you will be able to compare your tree to that of other studies.
Line 195: species should not be in italics.
Line 203-206: You will need to alter all this information based on the outcome of your phylogenetics when you include more systematically characterised Microsporidia. The genus Cucumispora was described by Ovcharenko et al. 2010, and further added to by Bojko et al. 2015 and Bojko et al. 2017. Bojko et al. 2017 explored the novel members of this genus using their sequence data as a proxy for taxonomy. In fact, before SSU information was gathered for Cucumispora dikerogammari, the species was named Nosema dikerogammari. Please read Ovcharenko et al. 2010 to see the details and other references.
Line 212: I want to remind you not to use Microsporidium as a group. It is a holding genus for many taxonomically diverse sequence isolates that need full characterisation. This is also mentioned in Bojko et al. 2016 – I strongly recommend that you read the introduction and phylogenetics of this paper.
Line 215: How can an outbreak be accidental? Was it accidentally introduced? Or is it a rare event? If so, how do you know it is rare… there is no continued monitoring of this species and its parasites.
Line 217-219: How can you conclude this is significant if you do not provide any statistical analysis of your data. Also, you pool many of your samples so you cannot give a reasonable estimate of parasite prevalence amongst specific populations, but may be able to provide an indication of species diversity.
Line 220-222: But you state earlier that all the sequences were different? Please be very clear about this.
Line 223: genus Cucumispora. But no, you have found similar sequences – these cannot be added to the genus officially without pathology and lifecycle information at minimum. If they are similar please give the detailed information such as coverage, sequence similarity, e value. This should also be reported in the results for every isolate you find, or at least in a suppl. table.
Line 229: This will be a specific isolate named Microsporidium. Please give the accession number from NCBI and the detailed information such as coverage, sequence similarity, e value.
Line 239-241: This conclusion needs much stronger foundation. Once you address the points I have made above you will be able to draw a relevant and accurate conclusion. Weren’t some hosts the only hosts to have certain Microsporidia? You say this above…
Line 244: The sequence isolates that we have identified during our study…
Suppl. Data: You need to provide data on the number of samples you actually tested. You only give the number of pooled samples. You need to provide similarity data to other Microsporidia gene isolates in the suppl. data or in the text of your report. You also need to provide your accession numbers for your isolates in the suppl. data. The presence of these on your tree is not sufficient as quality can often cause them to be illegible.

My final response:
I would like to review this manuscript again after the above changes have been made and the data analysed. I think this is a very interesting paper with good potential, but it is not valid within the literature in its current state, including: several errors around Microsporidia taxonomy; missing sequence data in the phylogeny; and no visible statistical analysis of the diversity data that you have collected. Much of my own research takes place upon the Microsporidia of amphipods, and much of it is mentioned above – please feel free to use this as a guide to find further literature that will help you to better analyse and discuss your data in this manuscript. Once these edits are completed it will be good to see this manuscript again. I have opted for you to know who I am, so you are welcome to look me up if you have further questions that I can help with.

Reviewer 2 ·

Basic reporting

The manuscript is generally well written. I have some minor issues.

Line 217-219: Generally, by comparing the percentage of infected hemolymph pools of different amphipod species from chosen sampling points (Table S1), we can conclude that the parts of infected individuals can vary significantly.

I don't think I understand what the authors are saying here. What do they mean by 'parts of infected individuals'? It sounds like they mean different tissues. If so, how does the percentage of infected hemolymph pools from different amphipod species show this?

I think this section needs rewording to make sure the point is clear.

Line 249-251: However, search for microsporidia in amphipods from higher depths (200-1600 m) is still necessary to confirm the observed homogeneity in distribution of these parasites.

I think ‘search for microsporidia in amphipods at greater depths’ or ‘search for microsporidia in amphipods at a greater depth (200-1600 m)’ is probably more suitable.

Experimental design

The authors make considerable play of detecting the presence of microsporidia using the hemolymph

'However, most previous studies utilized the whole individuals or soft amphipod tissues for amplification of microsporidian DNA, which does not rule out the possibility of contamination. To avoid this problem, our group recently conducted a study using molecular genetic techniques (SSU rDNA sequencing) to detect microsporidia in circulatory system of several endemic amphipod species from Lake Baikal (Madyarova et al., 2015). Searching directly in the hemolymph minimizes the possibility of identifying SSU rDNA of microsporidia located on exoskeleton, in gut lumen or inside parasites of the amphipods and can guarantee that the found microsporidia are parasitic directly to the amphipod species studied.'

I have no issue with the authors using the hemolymph but the above statement does raise a question. Is there any evidence that using hemolymph versus soft tissue produces notable differences in infection categorisations? It would be possible to do both types of extraction using the same animals. Has this been done? I noticed that the recent paper that surveyed infection in Lake Baikal (Ironside and Wilkinson, 2017) removes exoskeleton and gut tissue before using the soft tissue for DNA extraction. So they are also taking steps to try and prevent contributions from the gut and exoskeleton. I tend to agree that taking the hemolymph will reduce the likelihood of contamination. However, is there evidence that it does in practice? Also, are there limitations to using the hemolymph? Different microsporidia vary in their tissue distribution within hosts. Could some microsporidia be found at high levels in the soft tissue but low levels in the hemolymph? Again, a comparison of infections in the hemolymph and soft tissues of the same animals (if this has ever been performed) would be informative.

Validity of the findings

The discussion is relatively brief. The introduction refers to several studies that have looked at microsporidian infections in amphipods from Lake Baikal using molecular methods. (Kuzmenkova et al. 2008, Smith et al., 2008, Madyarova et al., 2015, Ironside and Wilkinson, 2017). However, the findings are only discussed in the context of Madyarova et al (2015). How do the findings change/support the picture provided by these other studies (if they do at all)? The authors describe the recent Ironside and Wilkinson (2017) paper as ‘indicating frequent introductions of the parasites into Lake Baikal ecosystem and high homogeneity of microsporidia between species of endemic hosts in the coastal zone’. Do the findings of infections and various locations and depths fit with this picture? I think a few lines in the discussion might help a general reader understand the relevance of this study within its broader context.

---

## Round 0.2 · Minor Revisions

Dear Authors, Thank you for submitting the revised version of the manuscript. I'm delighted to say your manuscript has been accepted. I have indicated minor revisions as one of the reviewers has indicated you might want to make a very minor revision to your manuscript for it is published. Thanks again for submitting to PeerJ

·

Basic reporting

All good.

Experimental design

All good.

Validity of the findings

All good.

Additional comments

Re-review of Dimova et al.
The authors have done a great job of incorporating the new information and I only have a few minor comments to be addressed below.

Minor suggestions:
- The genus Dictyocoela has now been finally been fully systematically defined. This happened during your revisions and is published under:
Bacela-Spychalska, K., Wróblewski, P., Mamos, T., Grabowski, M., Rigaud, T., Wattier, R., ... & Ovcharenko, M. (2018). Europe-wide reassessment of Dictyocoela (Microsporidia) infecting native and invasive amphipods (Crustacea): molecular versus ultrastructural traits. Scientific Reports, 8(1), 8945.
- Line 92 and 94: Microsporidia needs a capital M.
- Line 241-242: This data should be represented in the results first and not just in supplementary information. These are interesting findings and it would be good for you to display them in the results of the main manuscript.

Reviewer 2 ·

Basic reporting

The manuscript has been improved in accordance with the comments of both reviewers. The authors have made a genuine attempt to answer questions and clarify their writing. They have also altered/added information where asked. The small errors in English and sentence structure have been corrected.

Experimental design

The research question is still well defined. More details on the methodology have been added in response the reviewer’s comments. I believe the description of the methodology sufficient to allow other researchers to interpret the results presented and replicate the work.

Validity of the findings

The conclusions drawn by the authors have now been qualified appropriately and the authors make it clear that more study is required to confirm the lack of correlation. I believe the study adds to the existing literature on microsporidia infection and provides more information about the diversity of microsporidian parasites in Lake Baikal.

---

## Round 0.3 · accepted · Accept

Dear Authors, thank you for completing those additional minor corrections. I'm delighted to now accept your manuscript and thank you again for publishing in PeerJ.

#